EMBO
*reports*

*Scientific Report*

# SNX18 regulates ATG9A trafficking from recycling endosomes by recruiting Dynamin-2

Kristiane Søreng[1], Michael J Munson[1], Christopher A Lamb[2], Gunnveig T Bjørndal[1], Serhiy Pankiv[1], Sven R Carlsson[3], Sharon A Tooze[2] (iD) & Anne Simonsen[1,*] (iD)

## Abstract

**Trafficking of mammalian ATG9A between the Golgi apparatus, endosomes and peripheral ATG9A compartments is important for autophagosome biogenesis. Here, we show that the membrane remodelling protein SNX18, previously identified as a positive regulator of autophagy, regulates ATG9A trafficking from recycling endosomes. ATG9A is recruited to SNX18-induced tubules generated from recycling endosomes and accumulates in juxtanuclear recycling endosomes in cells lacking SNX18. Binding of SNX18 to Dynamin-2 is important for ATG9A trafficking from recycling endosomes and for formation of ATG16L1- and WIPI2-positive autophagosome precursor membranes. We propose a model where upon autophagy induction, SNX18 recruits Dynamin-2 to induce budding of ATG9A and ATG16L1 containing membranes from recycling endosomes that traffic to sites of autophagosome formation.**

**Keywords** ATG9; autophagy; dynamin; recycling endosome; SNX18
**Subject Categories** Autophagy & Cell Death; Membrane & Intracellular Transport

## Introduction

Macroautophagy (hereafter referred to as autophagy) is a lysosomal degradation pathway that mediates the breakdown and recycling of several cellular components such as damaged organelles and protein aggregates in order to maintain cellular homeostasis [1]. The process involves formation of a double-membrane phagophore that expands to sequester cytoplasmic material before it closes to form an autophagosome, which further delivers the sequestered material to lysosomes for degradation. The degraded material is released back into the cytosol to be reused by the cell. Autophagy is upregulated to provide nutrients during starvation periods and is also important for clearance of damaged or dysfunctional cellular components, thereby protecting against invading pathogens, neurodegeneration and cancer [1].

The process of autophagy is tightly controlled by autophagy-related (ATG) proteins, which comprise the molecular machinery necessary for autophagosome formation. The exact molecular mechanisms involved in activation of autophagy in response to different cellular conditions and types of cargo are yet to be fully understood, but it is generally believed that induction of autophagy requires initial activation of the ULK1 (unc-51 like kinase) and the PIK3C3 (class III phosphatidylinositol-3-kinase) complexes, leading to production of phosphatidylinositol-3-phosphate (PI3P) in endoplasmic reticulum (ER)-associated phagophore precursor structures [2]. The PI3P-binding protein WIPI2 is recruited to the forming phagophore [3] where it further interacts with the ATG12–5-16L1 complex to allow conjugation of Atg8 homologues of the LC3 and GABARAP subfamilies to phosphatidylethanolamine (PE) in the autophagic membrane [4,5]. Mammalian ATG9A is the only conserved transmembrane core ATG protein and has been found to localise to the trans-Golgi network (TGN), plasma membrane and endosomes, including recycling endosomes [6–10]. ATG9A cycles between these compartments and the continuous trafficking of ATG9A are thought to generate an ATG9A reservoir ready to contribute to autophagosome biogenesis upon induction of autophagy [10–12].

The biogenesis of the phagophore is not fully understood, but it is generally believed to originate from specific omega-shaped domains in the ER called omegasomes [13–15], found in close proximity to ER–mitochondria contact sites [16,17]. The phagophore expands by receiving membrane input through vesicle transport or membrane contact sites from different membrane sources, including ER-exit sites (ERES) and the ER–Golgi intermediate compartment (ERGIC) [18–20], the Golgi apparatus [21,22], mitochondria [23,24] and the plasma membrane [21,22,25]. Recently, it has become evident that recycling endosomes also play an important role during autophagosome biogenesis and both ATG16L1 and ATG9A traffic from the plasma membrane to recycling endosomes on their way to sites of autophagosome formation [7–10,26]. We recently identified the PX-BAR-containing protein sorting nexin 18 (SNX18) as a

1   Deparment of Molecular Medicine, Institute of Basic Medical Sciences and Centre for Cancer Cell Reprogramming, Institute of Clinical Medicine, Faculty of Medicine, University of Oslo, Oslo, Norway
2   Molecular Cell Biology of Autophagy Group, Francis Crick Institute, London, UK
3   Department of Medical Biochemistry and Biophysics, Umeå University, Umeå, Sweden
    *Corresponding author. Tel: +47 22851110; E-mail: anne.simonsen@medisin.uio.no
    [The copyright line of this article was changed on 14 March 2018 after original online publication.]

positive regulator of autophagy [26]. SNX18 promotes formation of recycling endosome-derived tubules containing LC3 and ATG16L1, indicating it facilitates autophagosome formation by delivery of proteins and membranes for the growing phagophore. We here present evidence that binding of SNX18 to Dynamin-2 is important for generation of ATG9A- and ATG16L1-containing autophagosome precursors from recycling endosomes. Depletion of SNX18 leads to accumulation of ATG9A in juxtanuclear recycling endosomes, resulting in less ATG16L1- and WIPI2-positive autophagosome precursors and reduced autophagic flux. This phenotype was reversed by re-introducing wild-type SNX18, but not a mutant of SNX18 deficient in Dynamin-2 binding.

## Results and Discussion

### SNX18 regulates ATG9A trafficking through recycling endosomes

It is becoming evident that trafficking of ATG9A and ATG16L1 through recycling endosomes is necessary for their function in induction of autophagy [7,10,12]. As SNX18 promotes autophagy by generating TfR-positive tubules positive for ATG16L1 and GFP-LC3 [26], we asked whether SNX18 may also regulate trafficking of ATG9A. Indeed, endogenous ATG9A colocalised with endogenous SNX18 (Fig EV1A) and was detected along the membrane structures induced by overexpression of SNX18 in HEK293A cells (Fig 1A). The colocalisation of ATG9A with TfR increased upon induction of autophagy by starvation (Fig 1B and C), in line with previous data showing ATG9A in recycling endosomes and an interaction of ATG9A with TfR in these cells [12].

To study a possible role of SNX18 in trafficking of ATG9A to autophagic precursor membranes, we generated a HEK293A SNX18 knock-out (KO) cell line using CRISPR-Cas9 technology [27,28]. Cells were transfected with a CRISPR-Cas9 plasmid together with an RNA guide targeting an exon just downstream of the N-terminal SH3 domain of SNX18, causing a premature stop codon and no production of SNX18 protein (Fig 1G). Interestingly, ATG9A accumulated in TfR-positive juxtanuclear recycling endosomes in SNX18 KO cells (Fig 1D and E), with approximately 50% of the SNX18 KO cells showing juxtanuclear TfR-positive structures compared to

approximately 10% of WT cells (Fig 1F). There was no increase in the overall levels of ATG9A and TfR in the SNX18 KO cells (Figs 1G and EV1F–H). In line with our previous results obtained with siRNA-mediated silencing of SNX18 [26], autophagic flux was reduced in SNX18 KO cells compared to wild-type (WT) cells, as analysed by lipidation of LC3 (LC3-II) (Fig 1G and H), GFP-LC3 puncta formation (Fig EV1B and C), p62 degradation (Fig EV1D and E) and degradation of long-lived proteins (Fig 1I). In contrast, selective autophagy seems not affected by SNX18 KO, as analysed by turnover of mitochondria upon deferiprone (DFP)-induced mitophagy (Fig EV1F). Taken together, our results indicate that SNX18 is important for trafficking of ATG9A from juxtanuclear recycling endosomes.

It was recently proposed that similar to yeast Atg9, mammalian ATG9A traffics through the Golgi, endosomes, including recycling endosomes, and a more peripheral ATG9A compartment, creating a continuous pool of ATG9A that can promote autophagosome formation when necessary [11,12]. Depletion of SNX18 does not cause a complete block in autophagy. Compared to inhibition or depletion of core autophagy components, the autophagic flux is reduced approximately 20% in the SNX18 KO cells (Figs 1I, and EV1J and K). As membrane input from recycling endosomes to the forming autophagosome is only one of several possible membrane sources, we do not expect the effect of SNX18 depletion to be as strong as that of a core autophagy protein. Given that SNX18 affects ATG9A-vesicle formation and trafficking from recycling endosomes and not from the Golgi apparatus or the more peripheral ATG9A compartment, the effect of SNX18 may be more evident after a longer period of starvation than the 2–4 h used in our experiments.

### SNX18 is required for recruitment of ATG16L1 to WIPI2 structures

The impaired autophagy seen in SNX18 KO cells was further supported by confocal imaging analysis showing reduced colocalisation of the early autophagic markers ATG16L1 and WIPI2 in starved SNX18 KO cells compared to starved control cells (Fig 2A and B). The lack of WIPI2-ATG16L1 colocalisation is caused by a significant reduction of the number of ATG16L1 and WIPI2 puncta formed in response to starvation in the SNX18 KO cells compared to WT cells (Fig 2C and D). These data are in line with our previous studies,

**Figure 1.  SNX18 regulates ATG9A trafficking from recycling endosomes.**

A   HEK293A cells were transfected with myc-SNX18 for 17 h, then fixed and immunostained against myc and ATG9A before analysis by confocal microscopy. Scale bar = 10 μm.

B   HEK293A cells were starved or not for 2 h in EBSS before fixation and immunostaining against ATG9A and transferrin receptor (TfR). The cells were analysed by confocal microscopy. Arrowheads mark ATG9A- and TfR-positive structures. Scale bar = 10 μm.

C   The colocalisation of ATG9A and TfR from (B) was quantified from > 100 cells per condition with Zen software (Zeiss) and normalised to fed state (mean ± SEM, *n* = 6). *$P < 0.05$, by Student's *t*-test.

D   HEK293A Ctrl or SNX18 KO cells were starved for 2 h in EBSS, fixed and immunostained against ATG9A and TfR, before analysis by confocal microscopy. Scale bar = 10 μm.

E   The colocalisation of ATG9A and TfR observed in (D) was quantified from more than 100 cells per cell line with Zen software (Zeiss) and normalised to control cells (mean ± SEM, *n* = 7). *$P < 0.05$, by Student's *t*-test.

F   Cells were treated as in (D) and the juxtanuclear localisation of TfR quantified in WT and SNX18 KO cells from > 100 cells per cell line (mean ± SEM, *n* = 6). ***$P < 0.001$, by Student's *t*-test.

G   HEK293A SNX18 Ctrl or KO cells were starved or not in EBSS for 4 h ± 100 nM Bafilomycin A1 (BafA1) before cell lysis and Western blot analysis.

H   LC3 lipidation from (G) was quantified, and the graph shows the average level of LC3-II relative to actin normalised to Ctrl fed (mean ± SEM, *n* = 6). *$P < 0.05$ as determined by two-way ANOVA and Bonferroni post-test.

I   Long-lived protein degradation was measured in HEK293A SNX18 Ctrl or KO cells as the release of $^{14}$C-valine after 4 h of starvation ± 3-methyladenine (3MA). The starvation-induced autophagic degradation is quantified as the difference in proteolysis in starved cells ± 3MA and normalised to the degradation of control cells (mean ± SEM, *n* = 3). *$P < 0.05$, by Student's *t*-test.

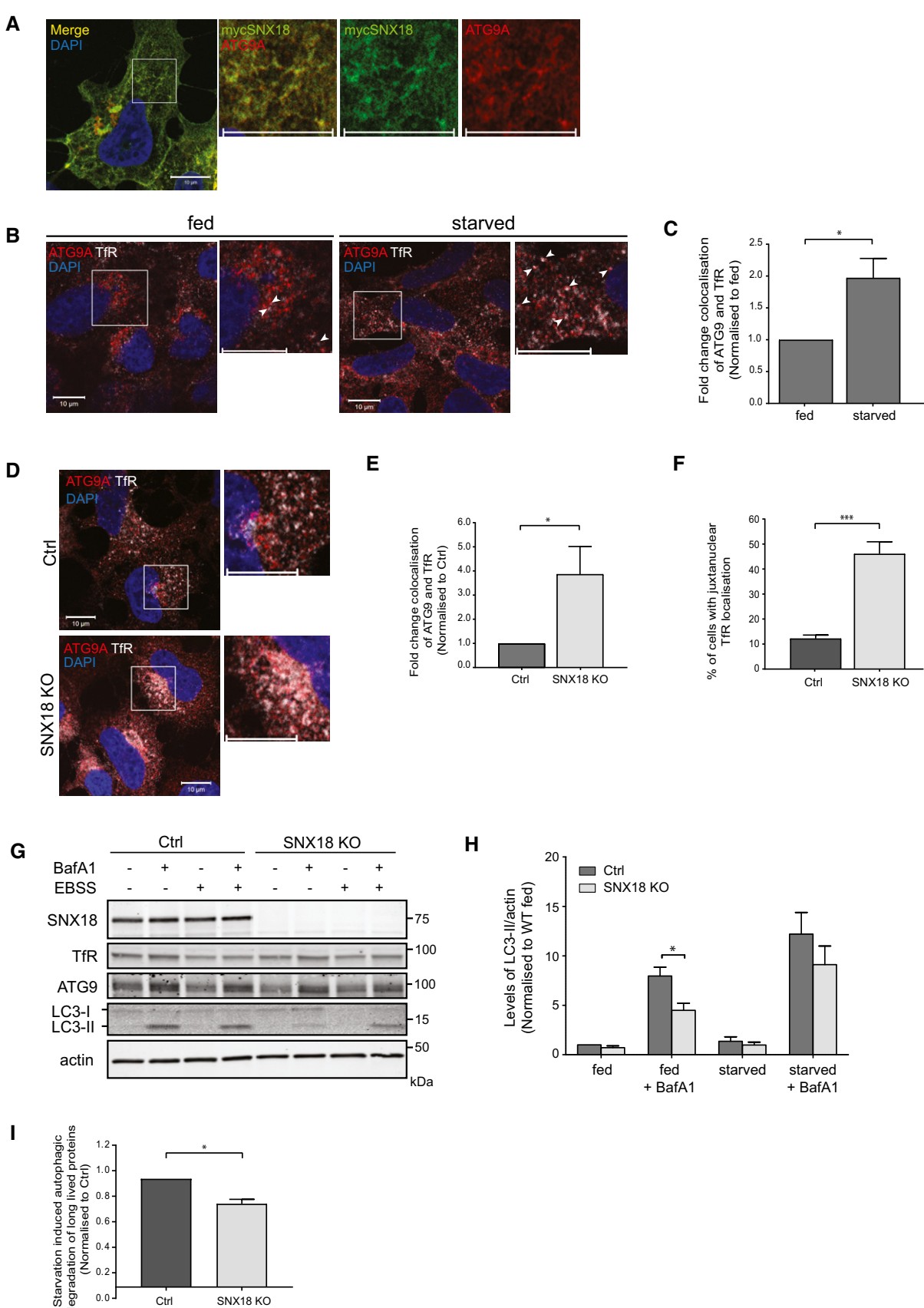

**Figure 1.**

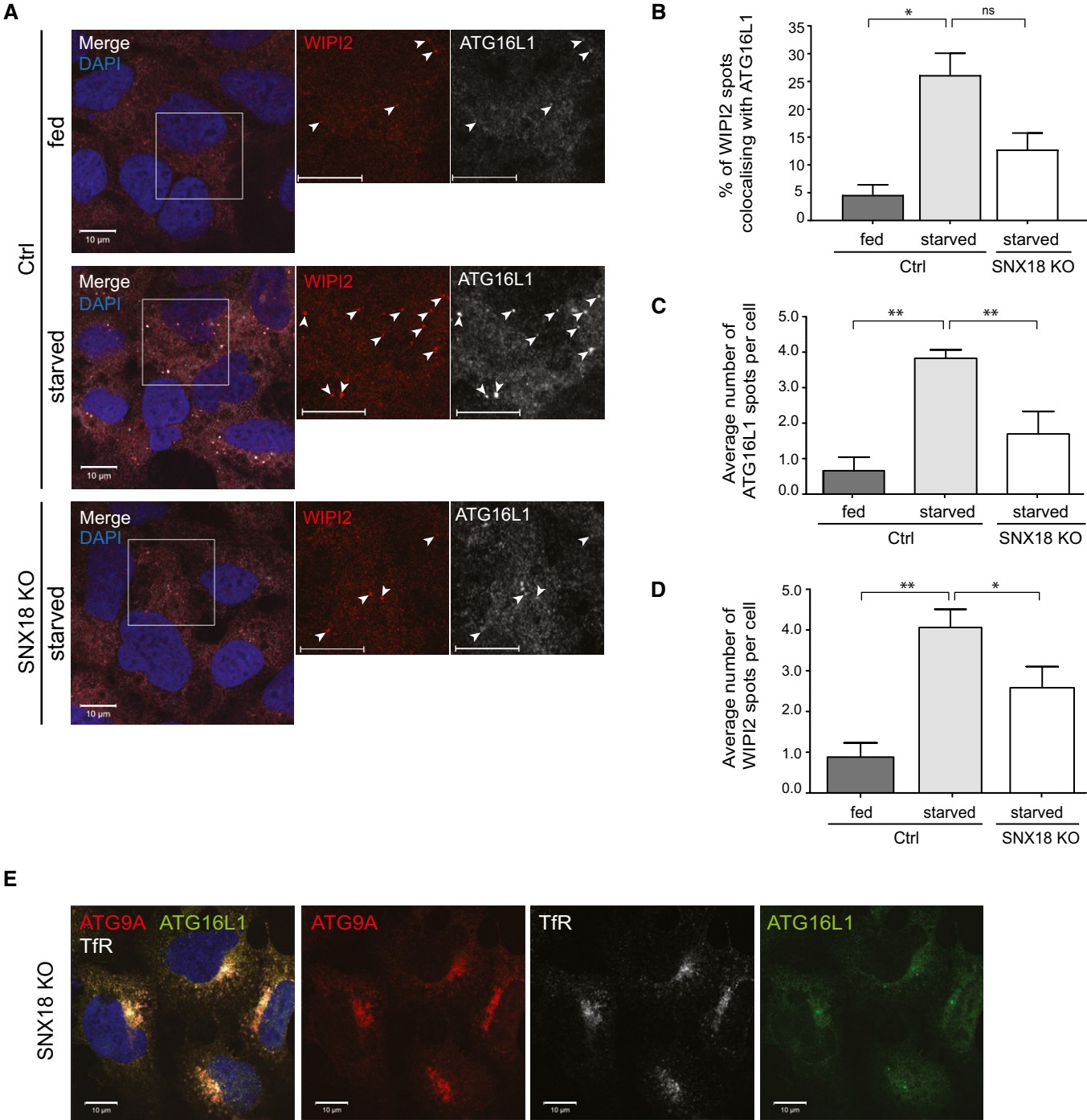

**Figure 2. SNX18 is required for recruitment of ATG16L1 to WIPI2 structures.**

A   HEK293A Ctrl or SNX18 KO cells were starved or not in EBSS for 2 h before fixation and immunostaining with antibodies against endogenous WIPI2 and ATG16L1. The cells were analysed by confocal microscopy. Arrowheads mark WIPI2- and ATG16L1-positive structures. Scale bar = 10 μm.

B   Cells were treated as in (A), and colocalisation between WIPI2 and ATG16L1 was quantified with Zen software (Zeiss) using 10 images per condition, including > 100 cells from three independent experiments. The graph shows the average colocalisation of WIPI2 and ATG16L1 (mean ± SEM, $n = 3$). Statistical significance was determined by one-way ANOVA and Bonferroni's multiple comparison test where *$P < 0.05$.

C   The number of ATG16L1 spots observed in (A) was quantified using CellProfiler software, and the graph shows the number of ATG16L1 spots per cell (mean ± SEM, $n = 3$). Significance was determined by one-way ANOVA and Bonferroni's multiple comparison test where **$P < 0.01$, *$P < 0.05$.

D   The number of WIPI2 spots observed in (A) was quantified as in (C).

E   HEK293A SNX18 KO cells were fixed and immunostained with antibodies against ATG16L1, ATG9A and TfR. Images were obtained by confocal microscopy. Scale bar = 10 μm.

showing that the starvation-induced colocalisation of ATG16L1 with the omegasome marker DFCP1 is reduced in SNX18-depleted cells [26]. In contrast, the number of WIPI2 spots was not reduced upon siRNA-mediated depletion of SNX18, which could be explained by insufficient knockdown efficiency. The reduced number of WIPI2 spots seen in SNX18 KO cells is likely due to the accumulation of ATG9A in juxtanuclear recycling endosomes, as dynamic ATG9A trafficking is important for formation of DFCP1- and WIPI2-positive omegasomes [12].

ATG9A and ATG16L1 have been found to traffic via the plasma membrane through recycling endosomes to the forming autophagosome [7]. Interestingly, ATG16L1 did not accumulate in the ATG9A- and TfR-positive juxtanuclear recycling endosome compartment seen in SNX18 KO cells (Fig 2E), suggesting that ATG16L1 could exit the recycling endosomes separately from ATG9A or that association of ATG16L1 with the recycling endosome membrane is SNX18 dependent. We have previously demonstrated a direct interaction between SNX18 and ATG16L1 [26], but further experiments and analysis of the relationship between ATG16L1 and SNX18 are needed to elucidate whether this interaction is responsible for the observed role of SNX18 in ATG16L1 recruitment to WIPI2-positive structures. A recent publication shows that lack of the sphingomyelin phosphodiesterase 1 (SMPD1) also leads to accumulation of ATG9A in juxtanuclear recycling endosomes without a corresponding accumulation of ATG16L1 [9]. This phenotype was associated with decreased colocalisation between ATG9A and WIPI2, but in contrast to our results, ATG16L1 was still recruited to WIPI2-positive structures.

Taken together, our results confirm that autophagy is reduced in SNX18 KO cells, similar to cells with siRNA-mediated depletion of SNX18, and further indicate that SNX18 is important for trafficking of the early core autophagy proteins ATG9A and ATG16L1 from recycling endosomes to sites of autophagosome formation.

## SNX18 and TBC1D14 have different functions at the recycling endosome

As SNX18 depletion results in juxtanuclear localisation of TfR-positive recycling endosomes, we asked whether SNX18 may influence the function of recycling endosomes. To test this, transferrin recycling was measured in cells depleted of or overexpressing GFP-SNX18. GFP-TBC1D14 was included as a control, as it has previously been shown that overexpression of TBC1D14 induces formation of tubules from recycling endosomes that inhibit both recycling endosome function and autophagosome formation [8]. Transfected cells were incubated with Alexa Fluor 647-transferrin (Tfn) for 15 min, followed by a chase at 15-min intervals before analysis of the remaining Tfn signal in the GFP-positive cells by flow cytometry. Whereas Tfn recycling was inhibited in GFP-TBC1D14-transfected cells, cells overexpressing SNX18 showed only a slight reduction in Tfn recycling (Fig EV2A). Moreover, siRNA-mediated silencing of SNX18 did not affect recycling of Tfn (Fig EV2B), in line with previous findings [29], suggesting that SNX18 is not required for the overall function of recycling endosomes. SNX18 was also not recruited to TBC1D14-induced recycling endosomes tubules (Fig EV2C).

TBC1D14 is a RAB11 effector protein and is known to regulate autophagy and ATG9A trafficking via binding to the TRAPP

complex [8,11]. It was found that proper ATG9A traffic between recycling endosomes and the Golgi apparatus depends on the generation of a tubular intermediate membrane compartment containing ATG9A, where a switch between RAB11 and RAB1 occurs and is facilitated by a TBC1D14-TRAPPIII-like complex. This change allows ATG9A to cycle from recycling endosomes back to the Golgi to maintain the ATG9A pool necessary for autophagy [11]. As SNX18 can facilitate membrane binding and tubulation through its PX-BAR domain, it remains to be investigated whether it contributes to the generation of the intermediate tubular compartment by cooperating with the TBC1D14-TRAPPIII-like complex. Interestingly, it was demonstrated that depletion of a subunit of the TRAPPIII-like complex resulted in accumulation of ATG9A on recycling endosomes [10], the same phenotype as observed in SNX18 KO cells. However, as SNX18 and TBC1D14 regulate autophagy oppositely and as SNX18 is not detected in TBC1D14-induced tubules, it is likely that SNX18 and TBC1D14 have different functions at the recycling endosome.

## SNX18 interacts with Dynamin-2 at ATG16L1-positive recycling endosome-derived membranes

The increased localisation of TfR- and ATG9A-positive membranes in the juxtanuclear region seen in SNX18-depleted cells suggests that SNX18 is needed for ATG9A trafficking away from this area. The GTPase Dynamin-2 has been identified as a binding partner of SNX18 [30,31] (Fig 3A) and since Dynamin-2 is involved in membrane fission during vesicle formation [32], we asked whether binding of Dynamin-2 to SNX18 is important for the function of SNX18 in autophagy. Indeed, endogenous Dynamin-2 was readily detected along the SNX18-positive tubular structures formed upon overexpression of mCherry-SNX18 in HEK293A cells (Fig 3B). Dynamin-2 interacts with SNX18 through binding of its proline-rich regions to the N-terminal SH3-domain of SNX18 [30] (Fig 3A). We generated a mutant version of SNX18 that could no longer bind Dynamin-2 by mutation of tryptophan 38 to lysine (W38K) in the SH3-domain of SNX18. This particular tryptophan is conserved in all SH3-domains and is crucial for binding to proline-rich regions [33,34]. By immunoprecipitation of myc-tagged SNX18 from HEK cell lysates, we found that Dynamin-2 interacts with myc-SNX18 WT, but not with the Dynamin-2 binding-deficient mutant myc-SNX18 W38K (Fig 3C). In contrast, the binding of SNX18 to endogenous ATG16L1, mediated by its PX-BAR domain, was not affected by the W38K mutant (Figs 3C and EV3F). In line with this, whereas Dynamin-2 was found to colocalise with membrane tubules formed upon expression of mCherry-SNX18 WT, Dynamin-2 was not detected on SNX18 W38K tubules (Fig 3B, D and E), although the total number of Dynamin-2 puncta was unaffected (Fig EV3A). Thus, our data indicate that SNX18 facilitates membrane recruitment of Dynamin-2.

To identify the membranes where the SNX18-Dynamin-2 interaction takes place, we took advantage of the split-YFP bimolecular fluorescence complementation assay [35]. In line with the IP results (Fig 3C), many YFP puncta were detected in cells expressing YFP1-Dynamin-2 and YFP2-SNX18 (or YFP1-SNX18 and YFP2-Dynamin-2) with only a few YFP spots seen in cells expressing the SNX18 W38K mutant (Fig 3F). The formed YFP puncta partially overlapped with TfR- and ATG16L1-positive structures (Fig 3G and H), indicating

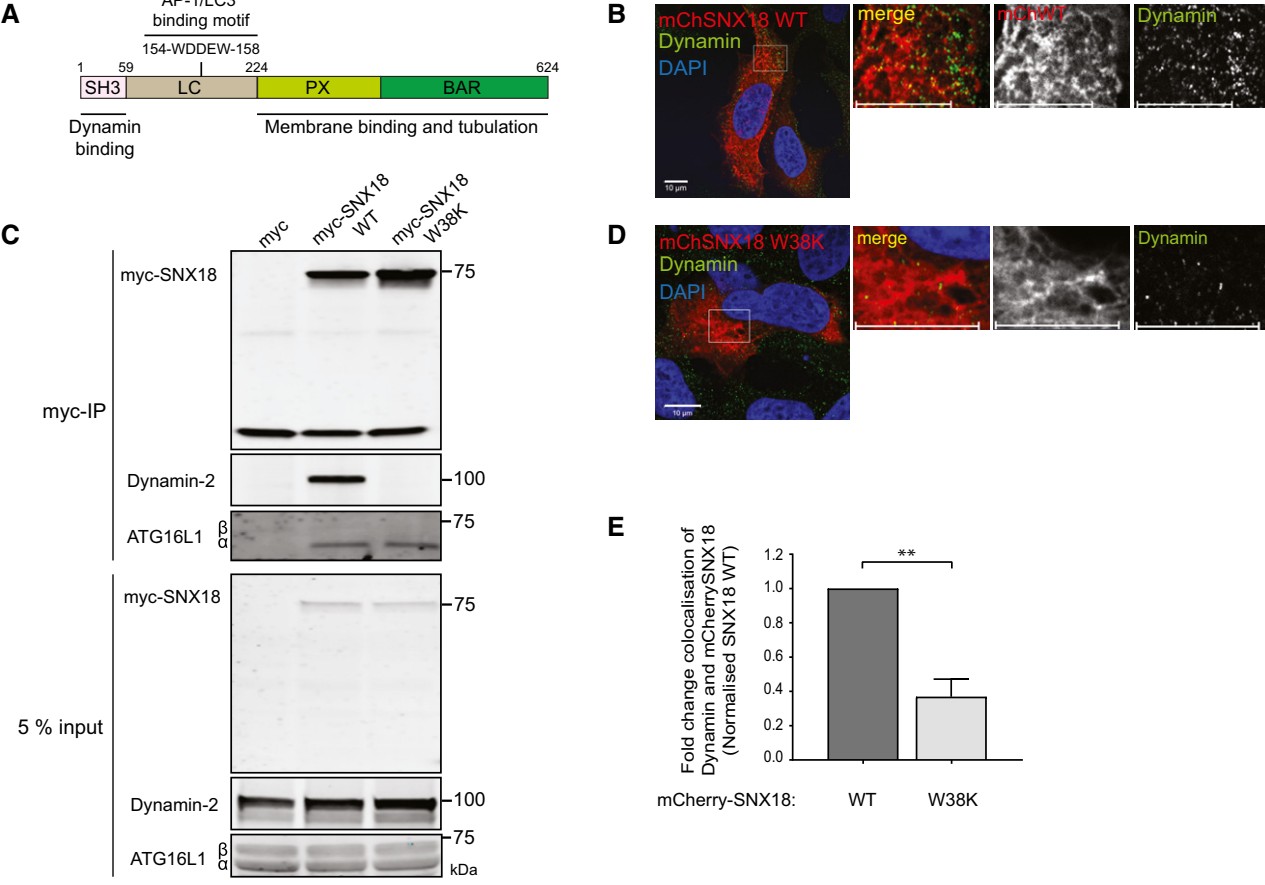

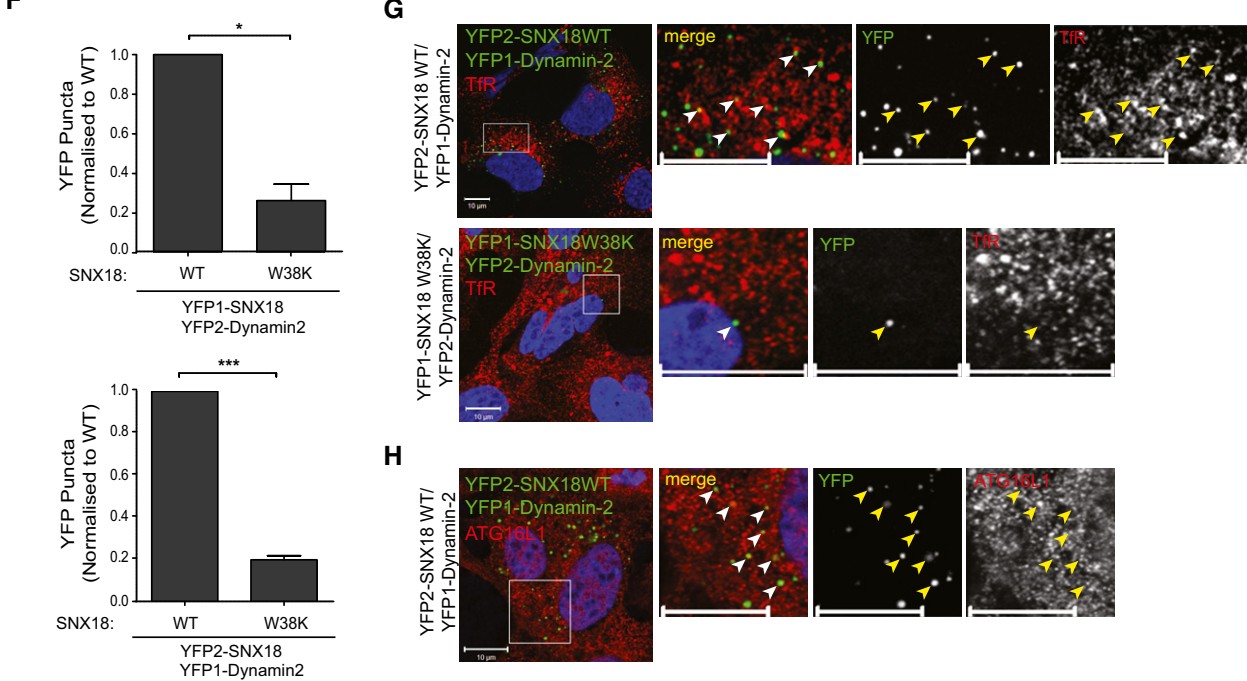

**Figure 3.**

**Figure 3.  SNX18 recruits Dynamin-2 to facilitate ATG9A traffic from recycling endosomes.**

A   The domain structure of SNX18 consists of an N-terminal SH3 domain followed by an unstructured LC region and a C-terminal PX-BAR domain responsible for membrane binding and tubulation. The SH3 domain binds the GTPase Dynamin-2, while the LC region binds to AP-1 and LC3/GABARAP.

B   HEK293A cells transfected with mCherry-SNX18 wild type (WT) were fixed and immunostained against Dynamin 17 h post-transfection. Images were obtained by confocal microscopy. Scale bar = 10 μm.

C   HEK293A cells were transfected with myc control, myc-tagged SNX18 WT or mutant SNX18 W38K before cell lysis. The lysates were incubated with magnetic anti-myc microbeads before immunoblotting of the cell lysate (input) and the immunoprecipitate (myc-IP) with antibodies against Dynamin-2, ATG16L1 and myc.

D   HEK293A cells transfected with mCherry-SNX18 W38K were fixed and immunostained against myc and Dynamin 17 h post-transfection and imaged by confocal microscopy. Scale bar = 10 μm.

E   The colocalisation of Dynamin and mCherry-SNX18 WT or W38K observed in (B) and (D) was quantified with Zen software (Zeiss) using 10 images per condition. The graph shows fold change colocalisation of Dynamin and mCherry-SNX18, normalised to cells expressing mCherry-SNX18 WT (mean ± SEM, $n$ = 3). **$P < 0.01$ by Student's $t$-test.

F   HEK293A SNX18 KO cells were cotransfected with YFP1-SNX18 WT or W38K and YFP2-Dynamin-2 or vice versa. Seventeen hours post-transfection, the cells were fixed and immunostained against TfR or ATG16L1 before analysis by confocal microscopy. The number of YFP spots was counted using CellProfiler software from 25 images per condition and normalised to SNX18 WT control (mean ± SEM, $n$ = 3). Significance was determined by Student's $t$-test where *$P < 0.05$ and ***$P < 0.001$.

G   The colocalisation of YFP puncta and TfR observed in the cells from (F) was analysed by confocal microscopy. Arrowheads mark YFP- and TfR-positive structures. Scale bar = 10 μm.

H   The colocalisation of YFP puncta and ATG16L1 observed in the cells from (F) was analysed by confocal microscopy. Arrowheads mark YFP- and ATG16L1-positive structures. Scale bar = 10 μm.

that the SNX18-Dynamin-2 interaction takes place at ATG16L1 containing recycling endosome membranes to promote membrane delivery to forming autophagosomes.

## SNX18 recruits Dynamin-2 to promote ATG9A trafficking from recycling endosomes

To investigate whether binding of SNX18 to Dynamin-2 is important for its role in trafficking of ATG9A from juxtanuclear recycling endosomes and autophagy, SNX18 KO cells were transfected with myc-tagged SNX18 WT or the Dynamin-2 binding-deficient SNX18 W38K mutant (Fig 4A). Indeed, the increased colocalisation of TfR and ATG9A observed in SNX18 KO cells (Fig 1E) was reversed in SNX18 KO cells expressing myc-SNX18 WT, but not upon expression of SNX18 W38K mutant (Fig 4A and B), indicating that ATG9A trafficking from recycling endosomes involves SNX18-mediated recruitment of Dynamin-2 to recycling endosome membranes to facilitate generation of vesicles from this compartment.

Not much is known about the regulation of ATG9A trafficking and its contribution to phagophore growth. It was recently shown that the N-terminal cytosolic tail of ATG9A harbours tyrosine- and di-leucine-based sorting motifs that interact with AP-1, AP-2 and AP-4 complexes [10,36,37]. Mutation of the AP2-binding motif inhibited trafficking of ATG9A from recycling endosomes and also autophagy [10], suggesting that AP-2-mediated trafficking of ATG9A from recycling endosomes is important for its role in autophagy. The interaction of ATG9A with the AP-1/2 complex was promoted by ULK1-mediated phosphorylation of ATG9A in response to starvation, leading to redistribution of mATG9A from the plasma membrane and juxtanuclear region to sites of autophagy initiation [36]. Furthermore, the RabGAP protein TBC1D5, which associates with ATG9A and the ULK1 complex during autophagy, was shown to mediate interaction with clathrin and the AP2 complex and proper sorting of ATG9A towards sites of autophagosome formation [38]. Recently, an interaction of ATG9A with AP-4 in the TGN was found to facilitate its transport to peripheral compartments and depletion of AP-4 to affect normal autophagosome formation [37]. SNX18 contains an AP-1 binding motif (WDDEW) in its unstructured LC region, which also mediates its interaction with LC3 and GABARAP proteins [26,30]. Here, we show that the SH3 domain of

SNX18 recruits Dynamin-2 to mediate formation of ATG9A vesicles from recycling endosomes, but whether or not this involves binding of ATG9 and/or SNX18 to any AP complex requires further investigation.

## Interaction of SNX18 with Dynamin-2 promotes autophagy

Since binding of SNX18 to Dynamin-2 is important for trafficking of ATG9A from juxtanuclear recycling endosomes, we asked whether this interaction also is essential for the function of SNX18 in autophagy. The degradation of long-lived proteins was measured in SNX18 KO cells transduced to stably express mCherry-SNX18 WT or mCherry-SNX18 W38K (Fig 4C). The autophagic flux, measured as the difference between starvation-induced proteolysis ± the autophagy inhibitor 3-methyladenine (3-MA), was significantly inhibited in the SNX18 KO cells compared to WT cells and could be rescued by SNX18 WT but not by the SNX18 W38K mutant (Fig 4D), indicating that SNX18-mediated membrane recruitment of Dynamin-2 promotes autophagy. The inhibitory effect of SNX18 KO and the W38K mutant on starvation-induced autophagic degradation is rather small (20% compared to inhibition of core autophagy components by 3MA), which is likely due to SNX18-mediated membrane delivery being only one of several sources of input to the forming autophagosome.

We next investigated whether expression of SNX18 was able to rescue the loss of ATG16L1 puncta and recruitment to starvation-induced WIPI2 spots seen in SNX18 KO cells (Fig 2A–C) and the importance of its binding to Dynamin-2. Whereas colocalisation between ATG16L1 and WIPI2 was readily detected in starved WT cells and in SNX18 KO cells with stable expression of mCherry-SNX18 WT, this was not the case in SNX18 KO cells and in KO cells expressing mCherry-SNX18 W38K (Fig 4E). Quantitation of the number of ATG16L puncta indeed demonstrated a partial but significant rescue with mCherry-SNX18 WT but not the W38K mutant (Fig 4F). Thus, efficient recruitment of ATG16L1 to WIPI2 spots is dependent on SNX18 and its binding to Dynamin-2.

SNX18 is a member of the SNX9 protein family of PX-BAR proteins. SNX9 interacts with Dynamin-2 and AP-2 to facilitate receptor-mediated endocytosis [39]. SNX9 and SNX18 were suggested to play redundant roles in endocytosis [31], and we therefore asked whether SNX9 might have a redundant role to SNX18 in

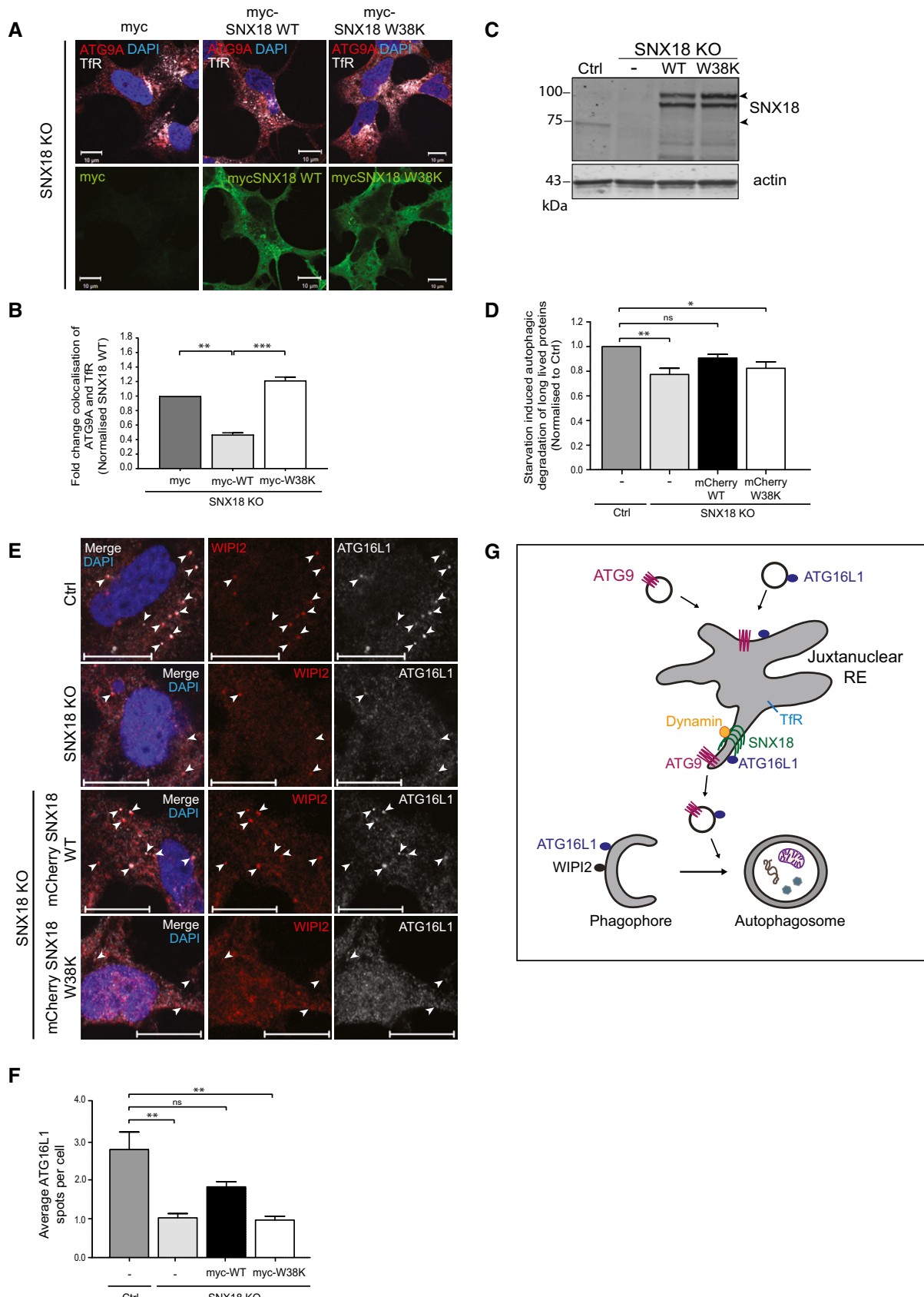

**Figure 4.**

◄

**Figure 4. SNX18-mediated recruitment of Dynamin-2 is important for its function in autophagy.**

A   HEK293A SNX18 KO cells transfected with myc control, myc-SNX18 WT or myc-SNX18 W38K were starved in EBSS for 2 h before fixation and immunostaining with antibodies against myc, ATG9A and TfR and analysed with confocal microscopy. The images show representative observations from SNX18 KO cells. Scale bar = 10 μm.

B   The colocalisation of ATG9 and TfR in the SNX18 KO cells from (A) was quantified with Zen software (Zeiss), from > 10 images per condition. The graph shows fold change colocalisation normalised to SNX18 KO cells transfected with myc control (mean ± SEM, *n* = 3). Significance was determined by one-way ANOVA and Bonferroni's multiple comparison test where **$P$ < 0.01, ***$P$ < 0.001.

C   Immunoblotting of SNX18 in cell lysates from HEK293A control, SNX18 KO and KO cells with stable expression of mCherry-SNX18 WT or -W38K mutant. Immunoblotting of actin serves as a loading control. Arrowheads indicate endogenous SNX18 (in WT cells) and mCherry-SNX18 WT and W38K.

D   Long-lived protein degradation was measured as release of $^{14}$C-valine after 4 h of starvation ± 3-methyladenine (3MA) in the cells shown in (C). The autophagic flux is quantified as the difference in proteolysis in starved cells ± 3MA and normalised to the flux in control cells (mean ± SEM, *n* = 3). Significance was determined by one-way ANOVA and Bonferroni's multiple comparison test where *$P$ < 0.05 and **$P$ < 0.01.

E   The cells shown in (C) were starved or not for 2 h with EBSS before fixation and immunostaining with antibodies against endogenous WIPI2 and ATG16L1. The cells were then analysed by confocal microscopy. The arrowheads mark WIPI2- and ATG16L1-positive structures. Scale bar = 10 μm.

F   Quantification of the number of ATG16L1 puncta from images represented in (E). Punctate structures were quantified using ImageJ and represented as the mean ATG16L1 puncta per cell ± SEM from *n* = 3 independent experiments. Significance was determined by one-way ANOVA and Bonferroni's multiple comparison test where **$P$ < 0.01.

G   Model for the role of SNX18 in ATG9A and ATG16L1 traffic. Trafficking of ATG9A and ATG16L1 through recycling endosomes is important for their function in autophagosome biogenesis. SNX18 promotes formation of ATG9A- and ATG16L1-positive vesicles from recycling endosomes, and its binding to Dynamin-2 is required for this. In cells lacking SNX18, ATG9A accumulates in juxtanuclear recycling endosomes and autophagy is inhibited. This can be rescued by WT SNX18, but not by a Dynamin-2 binding-deficient mutant SNX18, indicating that Dynamin-2 is involved in formation of ATG9A vesicles destined for the autophagosomes.

autophagy. SNX9 was not upregulated in SNX18 KO cells (Fig EV1H), and siRNA-mediated depletion of SNX9 did not inhibit starvation-induced autophagy or cause a further inhibition of autophagy in SNX18 KO cells, as analysed by LC3 immunoblotting (Fig EV3B and C) and ATG16L1 puncta formation (Fig EV3D and E).

In conclusion, we have demonstrated that ATG9A and ATG16L1 trafficking from recycling endosomes to sites of autophagosome formation are regulated by SNX18 and its binding to Dynamin-2. Dynamin-2 is recruited to recycling endosome-derived tubules induced by SNX18 through binding to the SH3-domain of SNX18. Through this mechanism, SNX18 promotes autophagosome formation. In the absence of SNX18, ATG9A accumulates in TfR-positive juxtanuclear recycling endosomes resulting in reduced autophagy (Fig 4G).

## Materials and Methods

### Antibodies and dyes

Antibodies used for immunoblotting are as follows: rabbit anti-SNX18 (Atlas antibodies, HPA037800, 1:500), rabbit anti-LC3B XP (Cell Signaling technology, 3868S, 1:1,000), rabbit anti-ATG9 (from Sharon Tooze [6], 1:1,000), mouse anti-TfR (Zymed, 13-6890, 1:1,000), mouse anti-β-actin (Cell Signaling technology, 3700, 1:5,000), rabbit anti-TBC1D14 (from Sharon Tooze [8], 1:500), rabbit anti-RAB11 (Invitrogen, 71-5300, 1:500), rabbit anti-Dynamin-2 (ABR/Thermo, PA1-661, 1:1,000), rabbit anti-SNX9 (from S. Carlsson [39], 1:1,000), mouse anti-p62 (BD biosciences, 610833, 1:1,000), rabbit anti-ULK1 (Santa Cruz, sc-33182, 1:250), mouse anti-ATG16L1 (MBL, M150-3, cl1F12, 1:1,000), mouse anti-flag (Sigma, F1804, 1:500). DyLight-680- and DyLight-800-conjugated secondary antibodies were obtained from Invitrogen (SA5-10170 and SA5-10044, 1:5,000). Antibodies used for immunofluorescence are as follows: mouse anti-WIPI2 (from Sharon Tooze [40], 1:2,000), rabbit anti-ATG16L1 (MBL, PM040, 1:200), hamster anti-ATG9A (from Sharon Tooze [41], 1:1,000), rabbit anti-ATG9A (Abcam, ab108338, 1:400), mouse anti-myc (IF and WB—DSHB, cl9E10, 1:20), rabbit

anti-TBC1D14 (from Sharon Tooze [8], 1:200), chicken anti-SNX18 (from Sven Carlsson, [26,30], 1:200), goat anti-myc (Abcam, ab9132, 1:500), mouse anti-TfR (Santa Cruz Biotechnology, sc-65877, 1:200), mouse anti-Dynamin (Upstate/Millipore, 05-319, 1:100). Alexa 488-conjugated secondary antibodies were obtained from Invitrogen (A-21206 and A-21202, 1:500), and Cy3 and Cy5-conjugated secondary antibodies were obtained from Jackson ImmunoResearch (Cy3: 715-165-151, 127-165-099 and 711-165-152, Cy5: 711-495-152 and 715-175-150). Alexa-647 transferrin was obtained from Invitrogen (T23366).

### Cell culture and reagents

HEK293A cells and HEK293A SNX18 control or knock-out cells were maintained in Dulbecco's modified Eagle's medium (DMEM; Lonza, BE12-604F) supplemented with 10% foetal bovine serum (FBS), 5 U/ml penicillin and 50 μg/ml streptomycin. For starvation in nutrient-deplete medium, the cells were incubated 2 h in Earle's balanced salt solution (EBSS; Gibco, 24010-043). Bafilomycin A1 (BafA1; Enzo, BML-CM110-0100) was used at 100 nM. 3-Methyladenine (3MA; Sigma-Aldrich, M9281) was used at 10 mM. Glass cover slips were coated with 20 μg/ml fibronectin (Sigma-Aldrich, F2006). Deferiprone (DFP; Sigma-Aldrich, 379409) was used at 1 mM. SNX18 WT/KO cells were stably transfected by lentiviral transduction of pLenti-III-PGK vectors. Briefly, HEK293FT cells were transfected with pLenti-III-PGK constructs in combination with pCMV-VSV-G and psPAX2 packaging vectors. Viral containing media was harvested 48 h post-transfection, passed through a 0.45-μm filter and added to SNX18 KO cells in the presence of 10 μg/ml polybrene (Santa Cruz, sc-134220). Transduced cells were selected by addition of 5 μg/ml Puromycin 24 h after addition of viral media.

### Transfection with siRNA or plasmids

HEK293A cells were transfected using Lipofectamine 3000 (Thermo Fischer Scientific, L3000015) for generation of SNX18 KO cells. For immunofluorescence, coimmunoprecipitation and Western blotting, plasmids were transfected using Lipofectamine 2000 (Invitrogen,

11668019). The following siRNA oligos were used: SNX18 (GCGGA GAAGUUCCCGGUCA), RAB11A (GUAGGUGCCUUAUUGGUUU), RAB11B (CAAGAGCGAUAUCGAGCUA) and ULK1 (UCACUGACC UGCUCCUUAA) obtained from Dharmacon and SNX9 (AAGAGAG UCAGCAAUCAUGUCU) obtained from Invitrogen. The siRNA oligos were delivered to the cells by Lipofectamine RNAiMAX (Invitrogen, 13778150) through reverse transfection.

## Plasmids

pCMV-myc-SNX18 has previously been published [30], whereas GFP and GFP-TBC1D14 are described in [8]. The flag-ATG16L1 construct used for IP has been previously described [42]. pDEST-myc-SNX18 WT and W38K were generated by subcloning of SNX18 from the pCMV-myc-SNX18 into the pENTR3C vector (Invitrogen) followed by site-directed mutagenesis of pENTR-SNX18 using the primers 5′-GGACATCGAGGGCAAGCTCGAGGGGGTC-3′ and 5′-GAC CCCCTCGAGCTTGCCCTCGATGTCC-3′ and finally LR cloning into the pDEST-myc vector (Gateway, Invitrogen). pDEST-EGFP-LC3B and pDEST-GFP-SNX18 have been generated in our laboratory by subcloning of SNX18 and LC3 into pENTR vector (Invitrogen) followed by LR cloning into pDEST-EGFP vector (Gateway, Invitrogen). pLenti-III-PGK mCherry SNX18 WT and W38K mutant were generated by Gibson assembly. For mitophagy analysis, a pLenti-III-PGK vector containing a mitochondrial localisation sequence with a C-terminal mCherry-GFP tandem tag was generated by Gibson assembly. pDEST-FlpIn-sYFP1-Dynamin-2 and pDEST-FlpIn-sYFP2-dynamin-2 were generated by subcloning of dynamin 2 from pcDNA3-HA-dynamin 2 (gift from Maria Lyngaas Torgersen/Kirsten Sandvig/Sandy Schmid, [43]) into pENTR3C vector (Invitrogen) using the primers 5′-TATAGGTACCATGGGCAACCGCGGGATG-3′ and 5′-TATACTCGAGCCTAGTCGAGCAGGGATGGC-3′ and LR Gateway cloning into the pDEST-FlpIn-sYFP1 or pDEST-FlpIn-sYFP2 vectors. pDEST-FlpIn-sYFP1-SNX18 WT and W38K and pDEST-FlpIn-sYFP2 WT and W38K were generated by LR Gateway cloning between corresponding pENTR-SNX18 WT or W38K and pDEST-FlpIn-sYFP1 or pDEST-FlpIn-sYFP2 vectors. pDEST-FlpIn-sYFP1 and pDest-FlpIn-sYFP2 vectors were generated by subcloning of Citrine (1–158) or Citrine(159–239) from pcDNA3-sYFP1 or pcDNA3-sYFP2 (gift from Veronika Reiterer-Farhan/Hesso Farhan [35]) using the primers for Citrine(1–158) 5′-CCGGAAGCTTATGGTGAGC-3′ and 5′-TATAGCTAGCCCCACCACCTCCAGAGC-3′ and for Citrine(159–239) 5′-CCGGAAGCTTATGAAGAACG-3′ and 5′-TATAGCTAGCCCCC ACCACCTCCAGAGC-3′ into HindIII and NheI sites of pDEST-FlpIn-HA vector instead of HA tag.

## CRISPR-Cas9

To generate SNX18 knock-out cells, three independent single-guide RNAs (#1: 5′-CACCCTACGCCAATGTGCCCCCCG-3′, #2: 5′-CACCGG CGCGGGCTTCCCGTACGG-3′, #3: 5′-CACCCTCGACGGCTCGTCTTC GGC-3′, targeting sequences are underscored) were designed against the 5′-end of human SNX18 downstream of the SNX18 SH3-domain using CRISPR Design Tool (crispr.mit.edu). The guides were cloned into the hSpCas9 (px330) plasmid (Addgene, 42230) and transfected into HEK293A cells. Transfected cells were isolated by serial dilution and expanded before the knock-out was confirmed by Western blotting. When using an affinity-purified antibody against the

SH3-domain of SNX18, no free SH3 domain could be detected by Western blotting in lysates from SNX18 knock-out cells (results not shown). The targeted regions in the genomic DNA were amplified by PCR and sequenced to identify the mutation leading to SNX18 knock-out. Guide #1 successfully knocked out SNX18 and was used in this study.

## Western blotting and immunoprecipitation

For confirmation of SNX18 KO and to monitor LC3 levels, cells were lysed in TNTE buffer (20 mM Tris pH 7.4, 150 mM NaCl, 5 mM EDTA, 1% Triton X-100) supplemented with complete protease inhibitor cocktail (Roche, 05056489001) for 5 min on ice. The lysates were centrifuged at $14,000 \times g$ for 10 min at 4°C to pellet the cell debris. The protein concentration of the supernatant was measured by BCA protein assay (Pierce, 23225) to ensure loading of equal amounts of protein on SDS–PAGE. Following SDS–PAGE, Western blotting was performed using primary antibodies and fluorophore-conjugated secondary antibodies for detection and analysis with the Odyssey Imaging System (LI-COR). For immunoprecipitation, cells were transfected with either myc, myc-SNX18 WT, W38K or flag-ATG16L1 and immunoprecipitated using the μMACS myc-isolation kit (Miltenyi Biotec, 130-091-123) or flag-isolation kit (Miltenyi Biotec) according to the manufacturer's instructions followed by Western blotting analysis.

## Immunofluorescence and confocal microscopy

Cells grown on fibronectin-coated glass cover slips were treated as described and then fixed in 4% paraformaldehyde (PFA; Poly-sciences, 18814-10) for 15 min on ice before quenching for 10 min in 0.05 M NH₄Cl and permeabilisation with 0.05% saponin in PBS for 5 min. Immunostaining was performed by incubating the cover slips with the indicated antibodies diluted in PBS with 0.05% saponin for 1 h at room temperature. The cells were counterstained for 10 min with 1 μg/ml Hoechst diluted in PBS and mounted in ProLong Diamond Antifade Mountant (Invitrogen, p36965) and analysed using a Zeiss 710 confocal microscope with a 63× objective lens. Colocalisation was measured using the Zen software from Zeiss where pixels from both channels are defined as colocalised if their intensities are above a set threshold. GFP-LC3, ATG16L1 and YFP puncta experiments were imaged using a Zeiss Axio Observer widefield microscope. Images were subsequently quantified for punctate structures using CellProfiler analysis software [44].

## Long-lived protein degradation assay

To measure degradation of long-lived proteins (being mainly degraded by autophagy), proteins were labelled with 0.25 μCi/ml L-[¹⁴C]-valine (Perkin Elmer, NEC291EU050UC) for 24 h in RPMI 1640 medium (Lonza, BE12-702F) containing 10% FBS. After labelling, the cells were washed in PBS and the ¹⁴C-valine was chased for 17 h in non-radioactive DMEM supplemented with 10% FBS and 10 mM L-valine (Sigma-Aldrich, V0513-25G) in order for short-lived proteins to be degraded. The cells were washed in EBSS and starved or not for 4 h in EBSS containing 10 mM L-valine and the presence or absence of 10 mM 3MA. The medium was collected, and the cells were washed with 1% BSA in PBS which was further added to the collected

medium. 50% Trichloroacetic acid (TCA) was added to precipitate radioactive proteins from the medium. The cells were lysed in 0.5 M KOH, and all the samples were transferred to counting vials containing 3 ml Ultima Gold LSC cocktail (Perkin Elmer, L8286). The radioactivity was measured by a liquid scintillation analyser (Perkin Elmer, Tri-Carb 3100TR), counting 3 min per sample, and the percentage degradation of long-lived proteins was determined by calculating the ratio of TCA-soluble radioactivity relative to the total radioactivity.

**Transferrin (Tfn) recycling by flow cytometry analysis**

To measure recycling of Alexa-647 Tfn, cells were transfected with GFP, GFP-SNX18 or GFP-TBC1D14. Twenty-four hours post-transfection, the cells were incubated in DMEM containing 10 μg/ml 647-Tfn for 15 min. The cells were then washed with PBS and chased for the indicated time periods in cell culture conditions at 37°C. At the end of the experiment, the cells were trypsinised, fixed with 4% PFA and centrifuged for 3 min at $1,000 \times g$ to pellet the cells. The cells were then washed in PBS and analysed by flow cytometry. The cells were separated into single cells and sorted according to the GFP signal. The amount of fluorescent Tfn in GFP-positive cells was measured using a flow cytometer (LSR Fortessa; BD).

**Statistical analysis**

Significance was determined using Graphpad Prism by one-way or two-way ANOVA, and Bonferroni's *post hoc* multiple comparison test or *P*-values were derived from two-tailed *t*-test for paired samples and considered statistically significant at $P \leq 0.05$ (see figure legends for further details).

**Expanded View** for this article is available online.

## Acknowledgements

We thank Toshiharu Fujita for help with the generation of CRISPR/Cas9 SNX18 KO cells. This work was partly supported by the Research Council of Norway through its Centres of Excellence funding scheme, project number 262652, the University of Oslo (KS and AS) and the Norwegian Cancer Society (MM and AS). CAL and SAT were supported by the Francis Crick Institute which receives its core funding from Cancer Research UK (FC001187), the UK medical research council (FC001187) and the Wellcome Trust (FC001187).

## Author contributions

KS, MJM, CAL, SP and GTB designed and performed the experimental research. KS drafted the article and made the figures. SAT and SRC provided essential reagents as well as preliminary results and revised the drafted article, and AS designed the project, analysed the data and wrote the final version of the manuscript.

## Conflict of interest

The authors declare that they have no conflict of interest.

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
