## [Review Process File · EMBO Reports]

SNX18 regulates ATG9A trafficking from recycling endosomes by recruiting Dynamin-2

Kristiane Søreng, Michael J. Munson, Christopher A. Lamb, Gunnveig T. Bjørndal, Serhiy Pankiv, Sven R. Carlsson, Sharon A. Tooze, Anne Simonsen

Review timeline:

Submission date:	14 July 2017
Editorial Decision:	11 August 2017
Revision received:	14 December 2017
Editorial Decision:	22 December 2017
Revision received:	4 January 2018
Accepted:	17 January 2018

Editor: Martina Rembold/Achim Breiling

Transaction Report:

1st Editorial Decision

11 August 2017

Thank you for the submission of your research manuscript to EMBO reports. We have now received reports from the three referees that were asked to evaluate your study, which can be found at the end of this email.

As you will see, all referees support the publication of your paper in EMBO reports. Nevertheless, referees #1 and #3 have a number of concerns and/or suggestions to improve the manuscript, which we ask you to address in a revised manuscript. As the reports are below, I will not detail them here, also as I think that all points should be addressed.

Given the constructive referee comments, we would like to invite you to revise your manuscript with the understanding that all referee concerns must be addressed in a point-by-point response. Acceptance of your manuscript will depend on a positive outcome of a second round of review. It is EMBO reports policy to allow a single round of revision only and acceptance or rejection of the manuscript will therefore depend on the completeness of your responses included in the next, final version of the manuscript.

I look forward to seeing a revised version of your manuscript when it is ready. Please let me know if you have questions or comments regarding the revision.

Referee #1:

The manuscript by Soreng et al. reports a role for SNX18 in the transport of ATG9 to the site of autophagosome formation. It was previously shown by the authors (Knaevelsrud, 2013, JCB (PMID: 23878278)) that SNX18 has a positive role in autophagy and regulates the trafficking of ATG16L1 from recycling endosomes (REs) to the autophagosome formation site. Here the authors extend

these findings to show that SNX18 contributes to the transport of ATG9 positive membranes from REs for autophagosome biogenesis. This activity depends on the previously reported dynamin binding activity of SNX18 (Park, 2010, JCS (PMID: 20427313)) and the authors propose a reasonable model according to which SNX18 dependent ATG9 and ATG16L1 transport carriers are formed at REs and undergo dynamin-mediated scission before they are recruited for autophagy. The study will be of interest for people working on membrane traffic and autophagosome biogenesis.

Specific comments:

1. The authors should specify if they are referring ATG9A or ATG9B.
2. The authors should determine the degree of co-localization between endogenous SNX18 and ATG9.
3. The block of autophagic flux by SNX18 KO as measured by LC3-II stabilization shown in Figure 1D is not convincing. Is the difference shown in the quantification (EV 1) significant?
4. For the data shown in Figure 1D, 1E and 4B it would be good to have a control such as ATG5KD, ATG16KD or wortmannin to see how large the effect of the SNX18 KO is compared to the depletion or inhibition core factors.
5. The authors should cite Park, 2010, JCS (PMID: 20427313) who showed that SNX18 interacts with dynamin.
6. As SNX18 and SNX9 were shown in the reference above to have redundant roles in endocytosis, it would be interesting to determine if co-depletion of these two proteins increases the autophagy phenotype seen by SNX18 depletion.
7. Further details on how the co-localization between WIPI2 and ATG16L1 (Figure 2A, B) was determined and quantified should be provided.
8. The effect seen for the W38K mutant on starvation induced autophagic degradation is very small (Figure 4B). The authors should discuss this more clearly.
9. Figure 3E should be quantified.
10. Figures 1B and 1F do not contain error bars.

Referee #2:

The authors have studied the role of the membrane remodelling protein SNX18 in ATG9 trafficking from recycling endosomes. They showed that ATG9 is recruited to SNX18-induced tubules generated from recycling endosomes. In cells deficient for the function of SNX18 they observed accumulation of ATG9 in juxtannuclear recycling endosomes. Along this process the authors identified that Dynamin-2 binding to SNX18 is important for ATG9 trafficking as well as for formation of autophagosomes. Thus, SNX18 by recruiting Dynamin-2 promotes budding of ATG9 and ATG16L1 containing membranes from recycling endosomes and autophagosome formation. The experiments are performed with all proper controls, conclusions are based on solid evidence, the manuscript is written in a balanced way and presents novel findings. Overall, this work is appropriate for EMBO R.

Referee #3:

The manuscript by Soreng et al., describes an investigation of the role of Snx18 in regulating the trafficking of ATG9, a membrane protein required for autophagy. Soreng and colleagues report that Snx18 binds to Dynamin 2 (Dyn2) and is required to recruit Dyn2 to the recycling endosome for the trafficking of ATG9 from recycling endosomes. Overall I think this manuscript has potential but there are some significant weaknesses that need to be addressed before it is suitable for publication.

Major points:

1. I'm unconvinced that Snx18 is genuinely responsible for recruiting Dyn2 as the manuscript states. Certainly Snx18 interacts with Dyn2 but the evidence that Dyn2 is recruited by Snx18 is weak. This area of the manuscript will need revision with extra data that more positively demonstrates the requirement for Snx18 to recruit Dyn2. The extent of colocalisation between Snx18 and Dyn2 is modest at best and although the Dyn2 staining appears less in cells expressing the W38K mutant, there is no apparent quantification of the localisation data. If Snx18 recruits Dyn2 and Dyn2 pinches off recycling endosome-derived tubules, would you not expect increased tubulation in cells expressing the Snx18 W38K mutant?
2. What was the rationale for making the W38K mutant? Why change a Trp to a Lys? Does a W38A mutant exhibit the same phenotype with respect to binding to Dyn2 and failing to rescue loss of Snx18 in the Snx18 KO cells? In my view, it is not at all clear whether the W38K mutant is folding properly, it certainly migrates differently to WT Snx18 on SDS-PAGE gels (see figure 3c) and the difference in migration is unexplained.
3. I think a western blot of ATG9 and TfnR in control and Snx18 KO cells would be helpful in figure 1 - are levels of these proteins increased or just their localisation altered? Additionally does loss of Snx18 result in reduced autophagic turnover of a classical substrate such as the poly-glutamine repeat fusion proteins often used as substrates.
4. In figure 2 where the fold change in colocalisation is shown, can the authors also show the actual numbers? If the colocalisation is modest to begin with, a pronounced fold change can be achieved with only a relatively small increase in actual colocalisation.
5. In figure 3F, how was the juxtannuclear localisation of the TfnR determined? Did it involve colocalisation with an accepted juxtannuclear marker? Was this experiment performed 'blind' so that bias was avoided when the cells were scored/assessed for juxtannuclear TfnR?
6. The myc-tagged Snx18 does appear to at least partially rescue the loss of Snx18 in the KO cells (figures 3F and 4D) but it would be helpful to see if it can rescue the LC3 phenotype as this is most directly associated with autophagy.

1st Revision - authors' response

14 December 2017

Referee #1:

The manuscript by Soreng et al. reports a role for SNX18 in the transport of ATG9 to the site of autophagosome formation. It was previously shown by the authors (Knaevelsrud, 2013, JCB (PMID: 23878278)) that SNX18 has a positive role in autophagy and regulates the trafficking of ATG16L1 from recycling endosomes (REs) to the autophagosome formation site. Here the authors extend these findings to show that SNX18 contributes to the transport of ATG9 positive membranes from REs for autophagosome biogenesis. This activity depends on the previously reported dynamin binding activity of SNX18 (Park, 2010, JCS (PMID: 20427313)) and the authors propose a reasonable model according to which SNX18 dependent ATG9 and ATG16L1 transport carriers are formed at REs and undergo dynamin-mediated scission before they are recruited for autophagy. The study will be of interest for people working on membrane traffic and autophagosome biogenesis.

Specific comments:

1. The authors should specify if they are referring ATG9A or ATG9B.

The antibodies we have used are specific to ATG9A. Moreover, all studies we cite have used ATG9A. We therefore use ATG9A instead of ATG9 throughout the manuscript. We have also updated the methods accordingly.

2. The authors should determine the degree of co-localization between endogenous SNX18 and ATG9.

In Fig 1A we show that myc-SNX18 colocalises extensively with endogenous ATG9A. We have previously shown colocalisation of endogenous SNX18 with endogenous ATG9 (see image below), but have unfortunately not been able to repeat the staining for endogenous SNX18. Previously we have used a chicken anti-SNX18 antibody to stain endogenous SNX18, but as the old batch of affinity-purified antibody was empty we had to affinity-purify more of this antibody. Unfortunately,

this new batch did not work. We have tried this antibody with four different secondary anti-chicken antibodies and have also included a positive control (chicken anti-myc). We have also tested again different rabbit polyclonal anti-SNX18 antibodies (home-made and from Atlas) without success. We here include an old image showing colocalisation of endogenous SNX18 and ATG9A, but as we are not able to quantify the data, we include it for the reviewer only.

3. The block of autophagic flux by SNX18 KO as measured by LC3-II stabilization shown in Fig 1D is not convincing. Is the difference shown in the quantification (EV 1) significant?

Knock-down or knock-out of SNX18 have a strong effect on formation of ATG16L1 and WIPI2 puncta (Fig 2), but do not completely inhibit LC3 lipidation or autophagic flux. The difference in LC3-II level is however reduced (significant in fed state, as quantified from 6 independent experiments) and we have now included the quantification of Fig 1D in the main figures (as new Fig1G and H).

4. For the data shown in Fig 1D, 1E and 4B it would be good to have a control such as ATG5KD, ATG16KD or wortmannin to see how large the effect of the SNX18 KO is compared to the depletion or inhibition core factors.

We have now repeated the LC3 blots in WT and SNX18 KO cells together with siRNA-mediated knock-down of ULK1 in WT cells (Fig EV1I and J). The level of LC3-II is significantly reduced, but not completely blocked, in siULK cells. It is difficult to compare this directly to the SNX18 KO cells (KD vs KO), but it is clear that depletion of a core autophagy machinery protein (e.g. ULK1) has a much stronger effect on LC3 lipidation than depletion of SNX18. As SNX18-mediated delivery of membrane from recycling endosomes to the forming autophagosome is only one of several possible membrane sources, we do not expect the effect of SNX18 depletion to be as strong as that of a core autophagy protein.

The LLPD (long-lived protein degradation) experiments (old Fig 1E and 4B, new Fig 1I and 4D) were already performed in the absence or presence of an autophagy inhibitor (3-methyladenine, 3MA) to measure the autophagic flux. The data presented represent the specific autophagic degradation, i.e. the difference in degradation in starved cells (WT or SNX18 KO/rescue) minus that in starved cells treated with 3MA. If we instead present the difference in % degradation in starved SNX18 KO cells vs starved WT cells treated with 3-MA, we find that the starvation-induced degradation in SNX18 cells KO is inhibited 20% compared to 3MA treated control cells.

5. The authors should cite Park, 2010, JCS (PMID: 20427313) who showed that SNX18 interacts with dynamin.

This paper has now been cited (page 8 and 12).

6. As SNX18 and SNX9 were shown in the reference above to have redundant roles in endocytosis, it would be interesting to determine if co-depletion of these two proteins increases the autophagy phenotype seen by SNX18 depletion.

We agree with the reviewer that it is important to check for a possible redundancy of these two proteins. We have previously shown that siRNA-mediated depletion of SNX9 alone has no effect on autophagy (Knævelsrud et al, J Cell Biol 2013), but it is of course possible that it would have a redundant function in SNX18 KO cells. As can be seen in Fig EV3, siRNA-mediated depletion of SNX9 in SNX18 KO cells (or in WT cells) had no further effect on the autophagy phenotype seen in SNX18 KO cells, as analyzed by LC3 western blot (Fig EV3B and C) and imaging of ATG16L1

puncta formation (Fig EV3D and E). Using qPCR, we also demonstrated that there is no significant upregulation of SNX9 at the transcriptional level (Fig EV1G).

7. Further details on how the co-localization between WIPI2 and ATG16L1 (Fig 2A, B) was determined and quantified should be provided.

The co-localization between WIPI2 and ATG16L1 was quantified using the Zeiss Zen software. 10-11 images of each condition from 3 independent experiments were used for quantification. This information is now included in the figure legend and the methods section.

8. The effect seen for the W38K mutant on starvation induced autophagic degradation is very small (Fig 4B). The authors should discuss this more clearly.

We agree that the effect of the W38K mutation is not so large (although significant) using the LLPD assay (old Fig 4B, new Fig 4D). Importantly, the degradation rate in the cells rescued with the mutant is similar to SNX18 KO cells and significantly different from control cells and SNX18 KO cells rescued with SNX18 WT. Overall, depletion of SNX18 (or expression of the dynamin binding-deficient mutant) causes a 20% reduction of autophagic degradation compared to cells treated with 3MA or depleted of ULK1. We believe this is because SNX18-mediated delivery of membrane from recycling endosomes to the forming autophagosome is only one of several possible membrane sources, and have now included a discussion of this in the revised manuscript (page 5). We also show that the colocalisation of ATG9A with TfR is increased in cells expressing the W38K mutant compared to WT SNX18 (Fig 4B) and that starvation-induced ATG16L1 puncta formation is inhibited (Fig 4F), indicating that trafficking of ATG9A and ATG16L1 from recycling endosomes is inhibited in cells expressing the dynamin-binding mutant.

9. Fig 3E should be quantified.

The data originally shown in Fig 3E has now been moved to Fig 4A and quantification of the data is shown in new Fig 4B.

10. Figs 1B and 1F do not contain error bars.

We apologize for this, which was due to the quantifications being done from only one experiment in the original submission. We have now included quantifications from three independent experiments and thus also error bars in Figs 1C and 1F.

Referee #2:

The authors have studied the role of the membrane remodelling protein SNX18 in ATG9 trafficking from recycling endosomes. They showed that ATG9 is recruited to SNX18-induced tubules generated from recycling endosomes. In cells deficient for the function of SNX18 they observed accumulation of ATG9 in juxtannuclear recycling endosomes. Along this process the authors identified that Dynamin-2 binding to SNX18 is important for ATG9 trafficking as well as for formation of autophagosomes. Thus, SNX18 by recruiting Dynamin-2 promotes budding of ATG9 and ATG16L1 containing membranes from recycling endosomes and autophagosome formation. The experiments are performed with all proper controls, conclusions are based on solid evidence, the manuscript is written in a balanced way and presents novel findings. Overall, this work is appropriate for EMBO R.

We thank the referee for these positive comments.

Referee #3:

The manuscript by Soreng et al., describes an investigation of the role of Snx18 in regulating the trafficking of ATG9, a membrane protein required for autophagy. Soreng and colleagues report that Snx18 binds to Dynamin 2 (Dyn2) and is required to recruit Dyn2 to the recycling endosome for the trafficking of ATG9 from recycling endosomes. Overall I think this manuscript has potential but there are some significant weaknesses that need to be addressed before it is suitable for publication.

Major points:

1. I'm unconvinced that Snx18 is genuinely responsible for recruiting Dyn2 as the manuscript states. Certainly Snx18 interacts with Dyn2 but the evidence that Dyn2 is recruited by Snx18 is weak. This area of the manuscript will need revision with extra data that more positively demonstrates the requirement for Snx18 to recruit Dyn2. The extent of colocalisation between Snx18 and Dyn2 is modest at best and although the Dyn2 staining appears less in cells expressing the W38K mutant, there is no apparent quantification of the localisation data. If Snx18 recruits Dyn2 and Dyn2 pinches off recycling endosome-derived tubules, would you not expect increased tubulation in cells expressing the Snx18 W38K mutant?

We agree with the reviewer that it is important to show that SNX18 indeed facilitates membrane recruitment of Dynamin-2 to autophagy-relevant membranes. We have now addressed this point in different ways. First, we have quantified the colocalisation of endogenous Dynamin-2 with SNX18 WT or W38K and find that the localisation of Dynamin-2 to SNX18 tubules is strongly reduced in cells expressing the SNX18 W38K mutant compared to WT SNX18 (Fig 3E). Importantly, the total number of Dynamin-2 dots were not affected (Fig EV3A).

To demonstrate where the Dynamin-2-SNX18 interaction takes place, we took advantage of the split-YFP system, where one part of YFP (YFP1) was fused to Dynamin-2 and the other part of YFP (YFP2) was fused to SNX18 WT or W38K mutant (or the other way around). A YFP signal is observed where there is an interaction between Dynamin-2 and SNX18. As can be seen in Fig 3F-G, many YFP puncta were detected in cells expressing YFP1-Dynamin-2 and YFP2-SNX18 (or YFP1-SNX18 and YFP2-Dynamin-2) while we hardly could detect any YFP spots in cells expressing the SNX18 W38K mutant. Interestingly, the YFP signal in cells expressing YFP2-SNX18 and YFP1-Dynamin-2 was partly overlapping with Transferrin receptor positive structures (Fig 3G), indicating the SNX18-Dynamin-2 interaction takes place at recycling endosome membranes. Moreover, when staining for endogenous ATG16L1 in the YFP2-SNX18 and YFP1-Dynamin-2 expressing cells, we could observe the YFP signal at or in close proximity to ATG16L1-positive structures (Fig 3H), indicating that SNX18 interacts with Dynamin-2 to facilitate transport of ATG16L1-positive membrane structures.

We agree with the reviewer that we might expect to see an increased tubulation in cells expressing the Snx18 W38K mutant if Snx18 recruits Dynamin-2 to pinch off recycling endosome-derived tubules. We do not however see a dramatic effect on the length of the mCherry-SNX18 W38K induced tubules compared to mCherry-SNX18 WT, but this is also very hard to quantify. We know from our previous study (Knævelsrud et al, J Cell Biol 2013) that the tubulation activity of SNX18 is tightly regulated by phosphorylation and dephosphorylation (of Ser-233) and believe that also the activity of the W38K mutant would be regulated in such a way.

2. What was the rationale for making the W38K mutant? Why change a Trp to a Lys? Does a W38A mutant exhibit the same phenotype with respect to binding to Dyn2 and failing to rescue loss of Snx18 in the Snx18 KO cells? In my view, it is not at all clear whether the W38K mutant is folding properly, it certainly migrates differently to WT Snx18 on SDS-PAGE gels (see Fig 3c) and the difference in migration is unexplained.

The W/Trp residue is conserved in all SH3 domains and it is well documented that it must be intact to allow binding of the SH3 domain to proline-rich regions. The W residue is solvent-exposed and has no interactions with main-chain atoms (sits in a shallow groove), thus mutation to any amino acid should not affect the folding (see e.g. Musacchio FEBS Lett 307:55 1992; Yu Science 258:1665 1992; Saksela and Permi, FEBS Letteres 2012). Mutation of W to K is the most commonly used mutation in the SH3-field since the mid-1990s (e.g. Tanaka MCB 15:6829 1995) and is why we chose this mutation, but several papers have shown that also mutation to an A has the same negative effect (e.g. Erpel EMBO J, 14:963 1995). The first reported inactivating mutation was W to R (v-Src) (Liu MCB 13:5225 1993).

We understand that the reviewer is concerned that the W38K mutant does not fold properly, as there is a shift in the migration of the W38K mutant compared to wild type SNX18 in the blot we included in the original submission (old Fig. 3C). We apologize for not explaining this shift in migration, which was due to the different vector backbones used to express myc-SNX18 WT (pCMV-myc-SNX18) and myc-SNX18 W38K (pDEST-myc-SNX18 W38K) in this particular blot, where myc-SNX18 W38K has a few extra amino acids between the myc-tag and SNX18 due to the Gateway system. We have now generated a pDEST-myc-SNX18 WT construct and show that this also interacts with Dynamin-2 and runs at the same MW as the W38K mutant (new Fig 3C).

As an additional control to show that the W38K mutation does not affect the folding of SNX18, we have now blotted for ATG16L1 in the IPs and show that endogenous ATG16L1 binds similarly to both the WT and W38K SNX18 (new Fig 3C). We have previously shown that ATG16L1 binds to the C-terminal PX-BAR region of SNX18 (Knævelsrud et al, JCB 2013) and therefore conclude that mutation of W38 to K does not affect the folding of SNX18. Binding of both SNX18 WT and W38K mutant to ATG16L1 was also demonstrated in an IP for Flag-ATG16L1 (Fig EV3F).

Because of the well documented effect (and use of) the W-to-K mutation and our additional control for proper folding (ATG16L1 IP) we did not feel that it was relevant to spend time and resources to make the W38A mutation in the SNX18 SH3 domain and repeat all experiments done with the W38K mutation. We hope the reviewer agrees.

3. I think a western blot of ATG9 and TfR in control and Snx18 KO cells would be helpful in Fig 1 - are levels of these proteins increased or just their localisation altered? Additionally does loss of Snx18 result in reduced autophagic turnover of a classical substrate such as the poly-glutamine repeat fusion proteins often used as substrates.

We have now included western blots for ATG9 and TfR in Fig 1G and quantification show that overall the levels of these proteins are not generally affected by SNX18 KO (quantifications shown in EV1F-H), although the levels of TfR are slightly increased.

We have also investigated the turnover of several classical substrates for selective autophagy, including p62 and damaged mitochondria. p62 is an autophagy receptor involved in selective autophagy, but is also itself an autophagic cargo upon starvation-induced autophagy. In line with a small reduction in starvation-induced autophagy in SNX18 KO cells, there is a small increase in the level of p62 in these cells (Fig EV1C-D). To address a possible role of SNX18 in mitophagy, control and SNX18 KO cells were transduced with lenti-virus to inducibly express a EGFP-mCherry-tagged mitochondrial protein, followed by induction of mitophagy by the iron-chelator deferiprone (DFP) and automated image analysis and quantification of red-only structures (mitochondria in lysosomes due to quenching of GFP in the acidic lysosomes) from several hundred cells. The level of DFP-induced mitophagy was not significantly changed in SNX18 KO cells (Fig EV1E).

4. In Fig 2 where the fold change in colocalisation is shown, can the authors also show the actual numbers? If the colocalisation is modest to begin with, a pronounced fold change can be achieved with only a relatively small increase in actual colocalisation.

We agree with the reviewer that this is an important point and have now included a graph showing the % colocalisation between ATG16L1 and WIPI2 (Fig 2B).

5. In Fig 3F, how was the juxtannuclear localisation of the TfR determined? Did it involve colocalisation with an accepted juxtannuclear marker? Was this experiment performed 'blind' so that bias was avoided when the cells were scored/assessed for juxtannuclear TfR?

We agree with the reviewer regarding the importance of blinding to avoid counting bias, unfortunately in the original submission this quantification was carried out in a non-blinded manner. The TfR juxtannuclear localisation was originally included as an approximate read out for defects in trafficking observed in the SNX18 KO and the ability to rescue these with SNX18 WT or W38K constructs. We feel that the accumulation of co-localisation between ATG9A and TfR upon SNX18 KO (seen in Fig1E) provides more detail than juxtannuclear TfR alone (in addition to being an unbiased quantitation method), however at the time of submission these experiments had not been completed in SNX18 rescue cell lines.

We have now completed these experiments and replaced the original Fig 3E with Fig4B. Expression of myc-SNX18 WT in SNX18 KO cells is able to rescue and reduce the ATG9A/TfR co-localisation (similar to control cells) but expression of myc-SNX18 W38K is similar to SNX18 KO cells. This therefore suggests that recruitment of Dynamin-2 by SNX18 is important for ATG9A trafficking from recycling endosomes.

6. The myc-tagged Snx18 does appear to at least partially rescue the loss of Snx18 in the KO cells (Figs 3F and 4D) but it would be helpful to see if it can rescue the LC3 phenotype as this is most directly associated with autophagy.

In this manuscript we show that SNX18 WT (but not the W38K mutant) is able to rescue the inhibition of autophagy seen in SNX18 KO cells, as analyzed by the degradation of long-lived proteins assay (Fig 4D), the starvation-induced induction of ATG16L1 spots (Fig 4E-F) and the

trafficking of ATG9 from recycling endosomes (Fig 4A-B). We agree that LC3 also is a good read-out for autophagy, but as the effect of SNX18 KO on LC3-II formation is less evident (approx. 20% reduction compared to depletion of core autophagy factors), we have not been able to obtain a significant rescue of the LC3 phenotype as analyzed by western blotting. We have however previously shown that myc-SNX18 WT is able to rescue the defect in starvation-induced GFP-LC3 spot formation seen in SNX18 siRNA depleted cells (Knævelsrud et al, J Cell Biol, 2013).

2nd Editorial Decision

22 December 2017

Thank you for the submission of your revised manuscript to EMBO reports. We have now received the full set of referee reports that is copied below.

As you will see both referees are positive about the study and support publication in EMBO reports without further revision.

Browsing through the manuscript myself, I noticed a few minor editorial changes that we need before we can proceed with the official acceptance of your study.

We look forward to seeing a final version of your manuscript as soon as possible. Please let me know if you have questions or comments regarding the revision.

Referee #1:

The authors have addressed most of my points adequately. It should be noted that the effects of SNX18 depletion on the process of autophagy appears to be very small. It is also unfortunate that the co-localization of endogenous SNX18 and ATG9A cannot be shown in this manuscript.

Referee #3:

The revisions have addressed the concerns I expressed previously and I am happy to recommend publication.

Corresponding Author Name: Anne Simonsen

Manuscript Number: EMBOR-2017-44837V1